# Culturing-Enriched Metabarcoding Analysis of the *Oryctes rhinoceros* Gut Microbiome

**DOI:** 10.3390/insects11110782

**Published:** 2020-11-11

**Authors:** Matan Shelomi, Ming-Ju Chen

**Affiliations:** 1Department of Entomology, National Taiwan University; Taipei City 10617, Taiwan; 2Department of Animal Science and Technology, National Taiwan University, Taipei City 10673, Taiwan; cmj@ntu.edu.tw

**Keywords:** cellulase, rhinoceros beetles, digestive system, symbiosis, microbiome

## Abstract

**Simple Summary:**

The coconut rhinoceros beetle is a pest of palm trees, which may have symbiotic gut microbes that help it digest its food. These microbes may produce enzymes like cellulase, which have uses in human industry. If the microbes are essential for the beetle’s survival, then finding ways to attack the microbes could help fight the pest. We sampled microbes from the guts of larval beetles collected in coconut trees in southern Taiwan, and identified the microbes both by culturing and with molecular biology methods. We found several species of bacteria and a yeast, *Candida xylanolytica*, with potential digestive functions for the beetle. Some of these microbes had been reported in these beetles before while others are new. Broader surveys of the beetle microbiome are needed to determine whether or not they have a conserved microbiome.

**Abstract:**

Wood-feeding insects should have a source of enzymes like cellulases to digest their food. These enzymes can be produced by the insect, or by microbes living in the wood and/or inside the insect gut. The coconut rhinoceros beetle, *Oryctes rhinoceros*, is a pest whose digestive microbes are of considerable interest. This study describes the compartments of the *O. rhinoceros* gut and compares their microbiomes using culturing-enriched metabarcoding. Beetle larvae were collected from a coconut grove in southern Taiwan. Gut contents from the midgut and hindgut were plated on nutrient agar and selective carboxymethylcellulose agar plates. DNA was extracted from gut and fat body samples and 16S rDNA metabarcoding performed to identify unculturable bacteria. Cellulase activity tests were performed on gut fluids and microbe isolates. The midgut and hindgut both showed cellulolytic activity. *Bacillus cereus, Citrobacter koseri*, and the cellulolytic fungus *Candida xylanilytica* were cultured from both gut sections in most larvae. Metabarcoding did not find *Bacillus cereus*, and found that either *Citrobacter koseri* or *Paracoccus* sp. were the dominant gut microbes in any given larva. No significant differences were found between midgut and hindgut microbiomes. *Bacillus cereus* and *Citrobacter koseri* are common animal gut microbes frequently found in *Oryctes rhinoceros* studies while *Candida xylanilytica* and the uncultured *Paracoccus* sp. had not been identified in this insect before. Some or all of these may well have digestive functions for the beetle, and are most likely acquired from the diet, meaning they may be transient commensalists rather than obligate mutualists. Broader collection efforts and tests with antibiotics will resolve ambiguities in the beetle–microbe interactions.

## 1. Introduction

Animals that feed on wood frequently depend on plant cell wall-degrading enzymes, such as cellulase and xylanase, to degrade their recalcitrant diet into digestible sugars. These enzymes can be produced by the animal endogenously, or by microbes living in the wood or in close external or internal symbioses with the host [1]. These two options are not mutually exclusive, as in the case of termites, where the insect and their gut microbes produce a cocktail of enzymes to fully digest wood into sugar monomers [2]. Interest in the digestive enzymes and microbial symbionts of wood-eating insects is growing due to demand for these enzymes in industries, such as bioconversion [3,4], on top of the basic science interests in how these systems differ across Insecta and how wood feeding convergently evolved in different groups [5,6].

The coconut rhinoceros beetle, *Oryctes rhinoceros* (Coleoptera: Scarabaeidae), is a widespread tropical pest of palm trees from India through the South Pacific [7]. The adults bore into and feed on the crown meristem, destroying the growing points, reducing yields, and stunting or killing the tree. They are pests of economically important species, such as coconut (*Cocos nucifera*), rattan (*Calamus* sp.), raffia (*Raphia ruffia*), areca nut (*Areca catechu*), oil palm (*Elaeis* sp.), pandan (*Pandanus* sp.), salak (*Salacca zalacca*), sago (*Metroxylon sagu*)*,* date (*Phoenix dactylifera*), and ornamental palms (e.g., *Roystonea regia*, *Livistona chinensis*, *Corypha umbraculifer*). They are also known to feed on non-palms, such as banana, taro, sugar cane, papaya, sugar apple, and pineapple [8,9]. While the adults are the destructive stage, the larvae feed on rotting wood from dead or fallen trunks. This wood is high in plant cell wall polysaccharides, requiring enzymes to break down, but also should be rich in cellulolytic microbes naturally found in decaying wood. As the larvae and adults do not interact, any symbiotic microbes the larvae need for digestion, if they exist, would either be passed down vertically and be housed in tissues, such as mycetocytes in the gut or fat body, or be acquired from the diet/environment. Analysis of the gut microbiome of such beetle larvae would reveal both transient microbes passing through the gut with the food and also symbionts, obligate or facultative, that reside in the gut longer than the transit time needed for food to pass or which replicate and develop in the gut to reach higher densities than in the substrate [6].

Several researchers have tried to characterize the microbiome of *Oryctes rhinoceros*, producing important findings like the pathogenic *Oryctes rhinoceros* nudivirus used in managing this pest [10] and the discovery of antimicrobial peptides such as rhinocerosin [11] and scarabaecin [12]. These efforts typically focused on bacteria and used either culturing or culturing-independent methods, such as metabarcoding and denatured gradient gel electrophoresis of the 16S rRNA genes to identify and quantify the microbial community in the guts, with only one study so far attempting both [13]. The results vary, with beetles in different locations having different microbes, and studies on other species of *Oryctes* showing different microbiomes yet [14,15]. Efforts to culture microbes expressing plant cell wall-degrading enzymes in particular have been successful but inconsistent. Cellulolytic *Bacillus brevis* was isolated from the beetles in Nigeria [16]; cellulolytic, xylanolytic, and mannolytic *Citrobacter koseri*, *Bacillus pumilus, Bacillus subtilis, Bacillus cereus,* and *Bacillus aryabhattai* were isolated in Java, Indonesia [17]; and cellulolytic *Citrobacter* sp. and lignolytic *Bacillus* sp. were isolated in Sumatra, Indonesia [18]. Species of *Citrobacter* and *Bacillus* are almost always found in *Oryctes rhinoceros* studies, though they are not always checked for enzymatic activity [13,19].

The goals of this experiment were to produce an in-depth look at the gut microbiome of *Oryctes rhinoceros* from a single location through a physiological context. This project sought to compare the microbiome in the larval midgut and hindgut with both culturing and culture-independent analysis, to look for microbial cellulases, and to describe the gut anatomy, which has not yet been done. This work thus provides a novel look at how gut physiology, enzymatic digestion, and microbiota interact in this pest beetle.

## 2. Materials and Methods

### 2.1. Insect Source

Larval *Oryctes rhinoceros* were collected from a single coconut (*Cocos nucifera*) grove on October 2019 in Tianyu village, Wandan Township, Pingtung County, Taiwan (22°35′05.4′′ N, 120°28′22.8′′ E). No approval was needed to collect or study these pest insects. *O. rhinoceros* is the only known *Oryctes* species in Taiwan, and larvae were identified as *Oryctes* using a key [20]. Insects and coconut pulp were collected in large plastic bins and transported to the Entomology Museum of National Taiwan University. Larvae survived entirely on the coconut palm they were transported with, and were maintained in the bins for up to two weeks until dissection. Larvae were between 50 and 70 mm in length at the time of dissection. Several larvae were maintained until adulthood, and were confirmed with keys and museum specimens to also be *O. rhinoceros*.

### 2.2. Dissection and Microbiology

Larvae were weighed, chilled in a freezer for a few minutes to briefly anaesthetize them, and surface sterilized for two minutes in 50% bleach and two rinses of ultrafiltered water, and then dissected under sterile conditions under a laminar flow hood. To culture microbes, incisions were made into the midgut and hindgut paunches (Figure 1) of six larvae and a weighed amount (≤500 mg) of contents was added to 1 mL of 10 mM phosphate buffer solution. Five serial 10-fold dilutions were spread onto plates of HiMedia^®^ Nutrient Agar (HiMedia Laboratories Pvt.Ltd., Dindhori, Nashik, India) and incubated aerobically at 30 °C for 72 h. Individual microbe colonies were isolated into pure cultures on the basis of morphology and counted across all plates with countable dilutions. The pH of the gut content was measured with a Hydrion^®^ pH strip (Micro Essential Laboratory Inc., Brooklyn, NY, USA) dipped into the lumen. Samples of gut contents were briefly examined under a compound microscope to observe microbes.

### 2.3. Cellulase Testing

To test the gut for cellulase activity, petri dishes were made with 50 mM citrate-phosphate buffer solution, 1% agar, and 0.2% carboxymethyl cellulose (CMC) (Fisher Scientific, Leicestershire, UK) or xylan from corn core (Tokyo Chemical Industry, Tokyo, Japan) as the only carbon source, with pHs of 5, 6.5, and 7.6. From a dissected larva, fresh gut luminal fluid was pipetted from incisions in the six gut sections as depicted in Figure 1 into small wells cut into the plates with a pipette tip, using 5 and 10 µL of fluid for CMC and xylan, respectively. Cellulase from *Asperillus niger* (Tokyo Chemical Industry, Tokyo, Japan) was used as a positive control, and purified water as a negative control. Plates were incubated at 40 °C in a plastic bag to retain moisture for 48 h. The plates were then flooded with 0.1% Congo Red (Acros Organics, Geel, Belgium) (Color Index No. 22120) in 0.1 M Tris-HCl buffer (Bioman Scientific Co., Ltd., Jhonghe City, Taiwan) pH 8.0 for 1 h on a shaking plate and then destained in 1 M NaCl (BioShop^®^ Canada Inc., Burlington, ON, Canada) for at least 1 h. Clear zones indicated cellulase activity.

To test for cellulolytic microbes, 50 µL from the initial 1-mL sample of midgut or hindgut gut contents as used for the nutrient agar plates were plated onto CMC agar at pH 5 and allowed to grow. After colonies were picked for isolation onto CMC agar, the plates were flooded with Congo Red and destained [21]. Halos of clearing in the plates qualitatively indicate extracellular cellulase activity. To detect intracellular cellulolytic activity for certain target microbes, the tetrazolium blue method for determining reducing sugars was used [22,23]. Microbes were grown in HiMedia^®^ nutrient broth at room temperature for 24 h, then diluted to match a McFarland 5 standard (Creative Media Plate^®^, New Taipei City, Taiwan) in optical density. This sample was diluted 1/100 in ultrafiltered water, and 10 µL of this dilution were mixed with 100 µL of 1% CMC in 100 mM sodium acetate buffer (pH 5.5), vortexed, and incubated at 37 °C for 13 min. To stop the reaction, 0.8 mL of tetrazolium blue reagent were added and the mixture boiled for five minutes. Absorbance at 470 nm was measured using a Libra S2 spectrophotometer (Biochrom Ltd., Cambridge, UK) and the reducing sugar concentration calculated by comparison with glucose solution standards. Broth without microbes was the negative control, and a cellulolytic strain of *Chaetomium globosum* (Sordariomycetes: Sordariales: Chaetomiaceae) maintained in the lab was the positive control.

### 2.4. Cultured Microbe Identification

Clones of individually cultured microbes were examined under a microscope, and a small isolated colony’s worth of cells lysed in 50 µL of nuclease-free water by heating at 95 °C for 10 min. Bacteria were identified using the 16S gene and fungi with the 18S, 26S (D1/D2), and internal transcribed spacer (ITS) (ITS1-5.8S-ITS2) genes using the primers in Appendix A. The PCR conditions were 95 °C for 3 min; 35 cycles of 95 °C for 45 s, 52 °C for 90 s, and 72 °C for 60 s; 72 °C for 10 min; and a hold at 4 °C. PCR products were run through an agarose gel and DNA extracted from the cut bands using a QIAquick^®^ Gel Extraction Kit (Qiagen, Hilden, Germany). The PCR products were sequenced at the Sanger DNA Sequencing Core Lab of the Center for Biotechnology at National Taiwan University with an Applied Biosystems^®^ 3730 analyzer (Thermo Fisher Scientific, Waltham, MA, USA). Forward and reverse sequences were merged using EMBOSS: merger [24] and identified using BLASTn and the 16S RNA database for bacterial sequences, or the appropriate fungal RNA database (ITS, 26S) for yeasts, searching only among type strain sequences [25]. For microbes whose identity required further validation, sequences for the respective barcoding genes from type strains of related species were downloaded from the DNA Data Bank of Japan (DDBJ) and the National Center for Biotechnology Information (NCBI) and used to make neighbor-joining trees. A representative strain of each microbe species was deposited at the Bioresource Collection and Resource Center (BCRC) in Hsinchu, Taiwan.

### 2.5. Metabarcoding

For complete bacterial microbiome analysis, DNA was extracted from 250 mg of midgut or hindgut contents from each individual larva using a Geneaid^®^ Presto™ Soil DNA kit (Geneaid Biotech Ltd., New Taipei City, Taiwan). This was done successfully (meaning, the extracted DNA passed quality to control for metabarcoding) for 16 larvae separately, including 4 of those from whose gut contents an aliquot was used for culturing. DNA was also successfully extracted from the fat bodies of four larvae, and from triplicate samples of coconut pulp from the larval rearing container. Quality control consisted of a NanoDrop (OD260/OD280) test, Agarose Gel Electrophoresis testing of DNA degradation and potential contamination, and Qubit 2.0 DNA concentration quantification. The extracted DNA was sequenced by BioTools Co. Ltd. (New Taipei City, Taiwan). Libraries were constructed on a paired-end Illumina HiSeq platform to generate 250-bp paired-end raw reads of the V3/V4 region of the 16S gene, which were merged using FLASH [26], and transformed to sequenced reads by base calling.

Results were analyzed via QIIME2 v2019.7 [27]. Denoising was done with *deblur*, discarding any features whose abundance in any sample was <1% of the average total reads for the samples. Samples with only one operational taxonomic unit (OTU) and OTUs present in only one sample were discarded. Sample microbial diversity was calculated with the minimum sampling depth to keep all samples (1553), which, for this data set, was still enough for the alpha rarefaction curves to plateau, meaning the sampling depth was sufficient to assess the species richness of the samples. OTUs were identified using the SILVA 138 99% OTUs Classifier for 16S [28,29]. The QIIME2 program *ancom* (analysis of composition of microbiome) was used to compare the microbiomes between the midgut and hindgut samples via differential abundance testing.

## 3. Results

### 3.1. Anatomy

A diagram of the larval digestive tract is present in Figure 1. The foregut is limited to a short pharynx or esophagus leading directly and with no obvious demarcation to a swollen cylindrical midgut. Two rings circle the midgut, each of approximately 30 cecae pointing posteriorly, approximately 1.5 mm long, and filled with a dark substance. The midgut narrows considerably before entering the hindgut, which starts with a bulbous swollen paunch. Notably, the gut forms an S-bend, with the “entrance” of the hindgut almost at the posterior end of the insect, while the “exit” of the hindgut is anterior nearer to where the midgut first thins. Thus, the contents of the hindgut are flowing in a posterior-to-anterior direction. The contents of both the midgut and the hindgut are dark brown with the consistency of fine wet soil. The hindgut exits into a thin “rectum” that exits through the anus. The mass of the hindgut is approximately 150% the mass of the midgut. The pH of the midgut varied considerably, averaging 6.67 ± 2.08, while the hindgut pH varied less drastically, averaging 6.92 ± 0.88. Neither section was consistently more or less acidic than the other. The rest of the body of the larva is comprised almost entirely of diffuse white fat body. Bacteria are visible in the midgut and hindgut contents, while sparse protozoa were visible in the hindgut.

### 3.2. Cellulase and Xylanase Activity of the Gut

Testing of the six gut sections identified in Figure 1 (midgut 1-3 and hindgut 4-6) plus a sample of soil in which the larvae lived showed compartmentalization of plant cell wall degradation was conducted (Figure 2). Cellulolytic activity was detected in all samples, including the soil, at pH 5 and 6.5, but limited to the hindgut paunch (Sections 4 and 5) at pH 7.6. Xylanolytic activity was limited to the hindgut paunch at pH 5, limited to the midgut at pH 6.5, and detected somewhat equally in Sections 1–5 at pH 7.6.

### 3.3. Cultured Microbes

The microbes found in the gut samples are listed in Table 1, with identifications based on sequence similarity to the type specimens in the 16S ribosomal RNA sequence database. The bacteria *Bacillus cereus* could be cultured from five out of six larval hindguts and three midguts. *Lysinibacillus fusiformis* was found in four hindguts and only one midgut. *Citrobacter koseri, Comamonas nitrativorans*, and an unidentifiable genus in the Enterobacteriaceae (with 97.74–97.75% 16S sequence similarity to *Salmonella enterica* subsp. *arizonae* strain DSM 9386, *Citrobacter youngae* strain GTC 1314, and *Enterobacter cloacae* strain DSM 30054) were found in the midguts and/or hindguts of three larvae. The former was the most abundant gut microbe in both the midgut and the hindgut by an order of magnitude. The other bacteria were only ever cultured from individual larvae. A fungus identified as *Candida xylanilytica* was found in five midgut and three hindgut samples. Because the genomic sequence of this cellulolytic *Candida xylanilytica* strain’s 18S, ITS (ITS1, 5.8S, and ITS2), and D1/D2 regions ambiguously identified it as either *Candida xylanilytica* or *Spathaspora* sp. (Ascomycota, Saccharomycetes, Debaryomycetaceae), neighbor-joining trees were made for those sequences and those of related species of *Candida* or *Spathaspora* to verify the identification. The results confirmed the cellulolytic yeasts isolated from the larval guts are strains of *Candida xylanilytica* (Figure 3, Appendix A).

Twelve microbe species could grow on CMC agar (Table 1)*,* including four strains, each found only in one larva, that were not found growing on nutrient agar plates. Only some *Candida xylanilytica* strains and the *Candida tropicalis* and *Streptomyces costaricanus/murinus* strains produced clearings in the agar after staining with Congo Red and destaining with NaCl. Seven *Candida xylanilytica* strains were tested for cellulolytic activity with the tetrazolium blue method, and all had cellulase activity (Table 2) regardless of whether they tested positive for extracellular cellulase activity with the Congo Red assays.

### 3.4. 16S Microbial Profiling

The raw metabarcoding reads are available on the NCBI Sequence Read Archive, GenBank Accession Numbers SRR11191467-SRR11191507. A total of 2,765,840 sequence reads were obtained in the raw data (min 31,370, max 86,727, mean 67,460). Following quality control, alpha rarefaction at several sampling depths using QIIME2 confirmed that our sequencing had reached a plateau for all samples, with plateaus reached at <200 sequences (Appendix A). QIIME2 identified 65 bacterial OTUs from the samples sent for metabarcoding, of which 33 were found in more than one sample (Appendix A). ANCOM analysis using QIIME2 revealed no significant differences between the larval midgut and hindgut microbiomes (W = 0 for all OTUs).

An OTU identified as *Paracoccus* was detected in fat body, hindgut, and especially midgut samples of most larvae and with higher abundance than any other OTU, though not found in the soil. *Citrobacter koseri* OTUs were similarly found in gut and fat body samples of several larvae and in high abundance, and not found in the soil. Notably, larvae with *Paracoccus* sp. as the dominant microbe almost always lacked *Citrobacter koseri*, and vice versa. *Lysinibacillus* was found in only three larval midguts, none of which were the ones also used for culturing. A putative new genus of *Anaeroplasma*-like Acholeplasmataceae was found in the midguts only of half the samples, and a putative new species of *Erysipelothrix* in the midguts of five individuals and fat body of one. A species of *Proteiniphilum* was found in four hindguts only but in low abundance. QIIME2 identified two OTUs as new genera of Oscillospiraceae, each in four separate hindguts, though these could possibly be a single species found in eight larval hindguts that QIIME2 mistakenly split. Common OTUs found in both midguts and hindguts were *Dysgonomonas* spp. in six samples and a new genus of Rhodocyclaceae in five. In contrast to the culturing results, no *Bacillus cereus* was found in the microbiome data, and the only Enterobacteriaceae was *Citrobacter koseri*.

## 4. Discussion

Microbes were cultured from a total of 6 larvae and metabarcoding from 16 larvae from the same coconut grove, which is a comparable sample size to those of similar studies that used typically no more than 5 samples [4,30,31,32,33,34,35], and is thus more than adequate to draw conclusions for beetles in this location. While culturing can only reveal a small portion of the microbial diversity of a sample, and aerobic culturing would miss the anaerobic microbes thought to dominate in an insect gut, the culturing data from this study is still fully comparable to culturing data performed by other researchers worldwide, and when combined with the metabarcoding can produce a more reliable picture of the true microbiome than either method alone.

*Bacillus cereus* is likely a ubiquitous microbial resident of the *Oryctes rhinoceros* digestive tract, as it or other members of the notoriously genetically similar *Bacillus* genus [36] have also been found in all previous studies of this species’ microbiomes [13,16,17,18,19]. While none of our strains produced cellulase with the Congo Red assay, cellulolytic strains of *B. cereus* exist, so it could provide symbiotic digestive functions to *Oryctes*. However, *B. cereus* is an extremely cosmopolitan species associated with a wide variety of animal digestive tracts [37,38,39], and so less likely to be an essential microbial symbiont for *O. rhinoceros* (unless one assumes that it plays important roles in an extremely diverse set of vertebrate and invertebrate organisms, which is indeed possible). More disconcerting is the absence of *B. cereus* from the molecular barcoding data; however, it is not unusual for the microbes found in culturing to differ from those in culture-independent data from the same individuals [40]. The alternative explanation is that *B. cereus* is a contaminant of the laboratory equipment involved in culturing but that is unlikely given its presence in all past *Oryctes rhinoceros* work from other laboratories [17,18,19]. Testing the *B. cereus* strains for extra- and intracellular cellulolytic activity would verify if they can help the larvae digest wood, and will be done in the future. Regardless of those results, these strains are almost certainly orally acquired and possibly transient gut microbes passing through the gut with the food.

*Citrobacter koseri* is the other microbe consistently found in *Oryctes rhinoceros* microbiome studies [13,17], and was the most abundant microbe in our samples in terms of CFU per mg, but it too is cosmopolitan and frequently found to be associated with insect digestive tracts [37]. Some strains of *Citrobacter koseri* and other Enterobacteriaceae can produce cellulases under certain conditions [41]. *Paracoccus* was found in the microbiomes of larvae that did not have *Citrobacter koseri*, and which were collected on a different day (Appendix A). An anaerobic cellulolytic *Paracoccus* species has been detected in a scarab beetle gut before [37]. Thus, while these microbes may well have digestive roles in *Oryctes rhinoceros*, not enough evidence exists at this time to claim this with certainty. Even if they are cellulolytic, these microbes are likely orally acquired and possibly transient gut microbes passing through the gut with the food, although none were found in the DNA extracted from the coconut pulp itself (Appendix A).

While several microbes grew on CMC agar, little evidence of extracellular cellulase production was found using the Congo Red assays except for some *Candida xylanilytica* strains. Testing for cellulase activity with the tetrazolium blue method was only performed for *Candida xylanilytica*, and yielded positive results. This recently discovered fungus [42] is related to the *Spathaspora* clade known to be associated with rotting wood and wood-boring insects [43], and likely provides enzymatic digestion for the beetle, but whether this function is essential to the fitness of the species is presently unknowable. This microbe was found in the midguts and hindguts of almost all beetles in this study but had not been noted in previous studies, largely because they focused on bacteria. Future collection efforts across Taiwan are underway to determine how widely spread this yeast is in *Oryctes rhinoceros* digestive tracts. Cellulolytic strain L1M1 of the *Candida xylanilytica* is stored and publicly available at the Bioresource Collection and Research Center in Hsinchu, Taiwan (BCRC: 23544). Because this metabarcoding study only looked at bacterial diversity, it missed the *Candida xylanolytica* along with any protozoa in the samples, some of which were visible in the hindgut samples under a microscope (albeit in far lower abundance compared to an equivalent sample from a termite). Future projects focusing on fungal and protozoan diversity in *Oryctes* are thus justified.

*Lysinibacillus fusiformis* (syn. *Lysinibacillus sphaericus, Bacillus fusiformis, Bacillus sphaericus*) was commonly found, yet is also cosmopolitan, and a known insect pathogen [44]. While cultured often, it was less abundant in the metagenomic data, suggesting it is not as widely conserved in the larval guts. The putative symbionts of *Proteiniphilum, Erysipelothrix,* and the unidentifiable Oscillospiraceae and Acholeplasmataceae are seemingly specific to either the midgut or the hindgut and could be linked to the different pH optimal for cellulolytic activity in the midgut and hindgut; however, they are not known from previous molecular datasets. Likewise, several microbes identified in previous work as being possible symbionts were not observed in this study: namely *Desulfovibrio* sp., *Treponema* sp., *Endomicrobium* sp., and *Bastocystis* sp. [13]. Comparative microbiome analysis of samples found across greater geographic distance and feeding on different plants are necessary to separate transient microbes picked up from the specific location where a larva feeds from genuine symbionts, and such work is presently ongoing. One should also remember that not all organisms have an obligate microbiome [45], and transient or environmentally acquired gut microbes can still have powerful impacts on insect survival and fitness, so final conclusions on the importance of microbes to an insect’s fitness will require assays with antibiotics or axenically reared insects.

The function of the two rings of midgut cecae remains unknown, but the data does not suggest a microbe-related hypothesis for their function. The midgut and hindgut are clearly separated from each other, and their gut fluids are cellulolytic at different pH levels, suggesting compartmentalized digestion [46]. The large paunch-like hindgut of the beetles and the greater cellulolytic and xylanolytic ability detected there suggest a microbial fermentation chamber, and the S-bend of the gut would increase the transit time for food to provide increased time for digestion [46]. However, the absence of significant differences in the pH or bacterial diversity of the midgut and hindgut in *Oryctes rhinoceros* cast doubt on whether microbial fermentation is critical for cellulase digestion in the gut the way it is in lower termites. Rather, the gut may be an opportune location for cellulolytic microbes found in the food to continue their activity, while, otherwise, the beetle’s digestion is symbiont independent, using the endogenous cellulase identified in the *Oryctes rhinoceros* transcriptome [13]. Microbe-independent digestion is surprisingly common in insects [45], so insects should not automatically be assumed to have symbionts. The lack of opportunities for horizontal microbe transfer in these non-colonial beetles would preclude the evolution of obligate symbioses like in the termite hindgut; however, the existence of vertically transmitted microbes cannot be ruled out at this time, and the hypothesis that cellulolytic *B. cereus* and *Candida xylanilytica* play digestive functions in the insect is still supportable. Molecular analysis of the microbiome, if present, of the adult ovaries would quickly identify potential vertically transmitted microbes, and fluorescent in situ hybridization of these species would better visualize in what gut areas or tissues these microbes are found [47].

## 5. Conclusions

The digestive system of the *Oryctes rhinoceros* larva includes a midgut with cecae and a swollen hindgut paunch. Both gut sections show cellulolytic and xylanolytic activity, and the bacterial populations in the two gut sections are similar. Dominant microbes include *Citrobacter koseri, Bacillus cereus,* an unculturable *Paracoccus* sp., and the fungus *Candida xylanolytica*. Some of these microbes are likely acquired from the environment given their known cosmopolitanism, but all have documented or demonstrated cellulolytic abilities, so whether they are commensalists or mutualists that help the beetle larvae digest their food remains to be proven.

## Figures and Tables

**Figure 1 insects-11-00782-f001:**
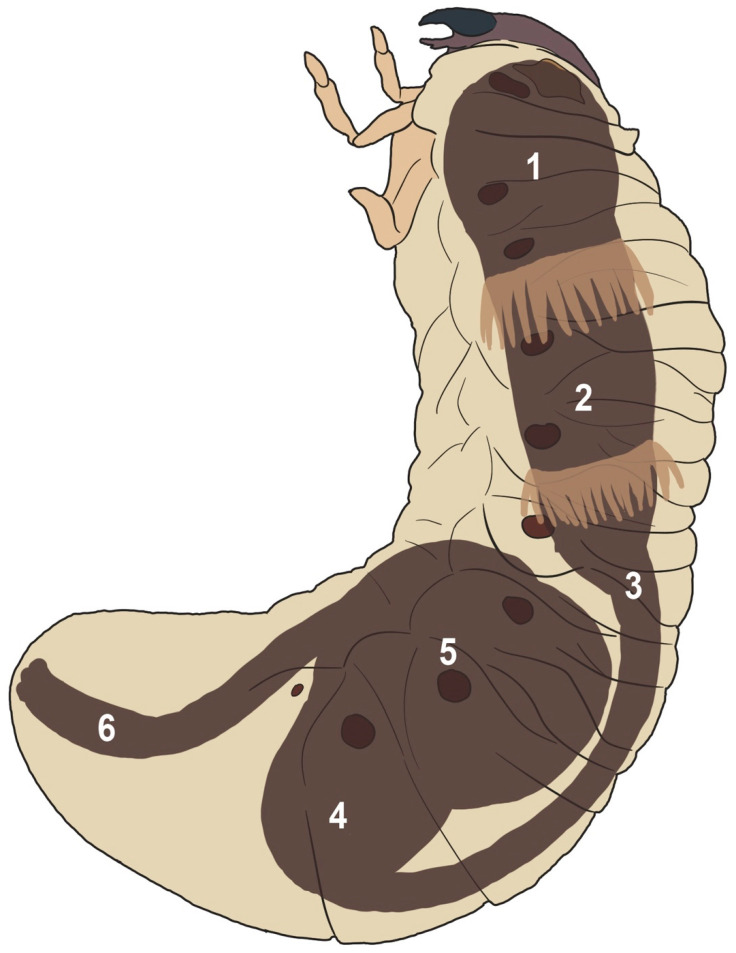
Diagram of the digestive tract of a larval *Oryctes rhinoceros*. The numbering corresponds to the cellulase and xylanase tests in Figure 2. The foregut is short to nonexistent. Two rows of gastric cecae ring the midgut (1–3). Our midgut samples were taken from an incision made between these rings (Section 3). The hindgut (4–6) contains a large bulbous fermentation chamber whose entrance from the midgut is posterior to its exit, meaning gut contents flow through the hindgut in a posterior-anterior direction. Our hindgut samples were taken from an incision in the middle of this paunch (5). Figure Credit: Sonja Pinck.

**Figure 2 insects-11-00782-f002:**
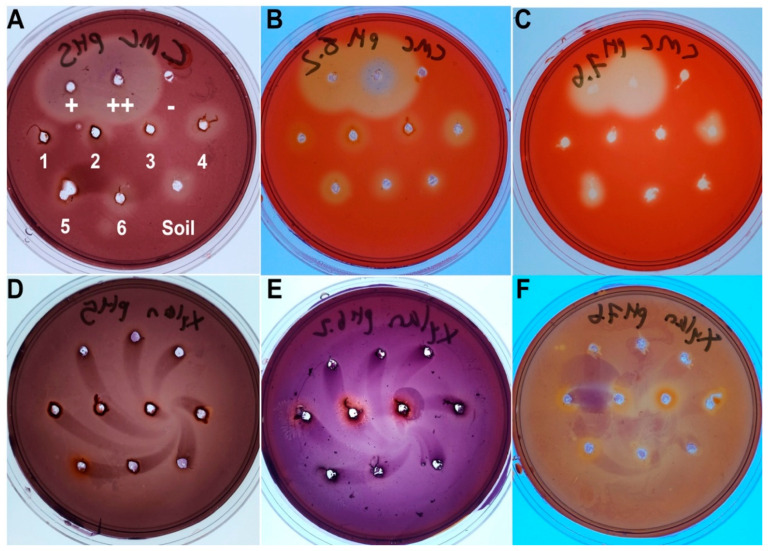
Cellulase and xylanase tests of the *Oryctes rhinoceros* gut at different pHs. Clearings in the Congo Red-stained plates indicate positive cellulase or xylanase activity. The positive control is commercial cellulase from *Asperillus niger* (TCI, Japan): (+) is 0.029 U, (++) is 0.29 U. Note that this cellulase is not xylanolytic. The negative control (-) is deionized water. The numbering corresponds to the sections in Figure 1, with 1–3 from the midgut and 4–6 from the hindgut. The other sample is from the soil in which the larvae live. (**A**) Carboxymethyl cellulose CMC plate at pH 5. (**B**) CMC plate at pH 6.5. (**C**) CMC plate at pH 7.6. (**D**) Xylan plate at pH 5. (**E**) Xylan plate at pH 6.5. (**F**) Xylan plate at 7.6. Photo Credit: Matan Shelomi.

**Figure 3 insects-11-00782-f003:**
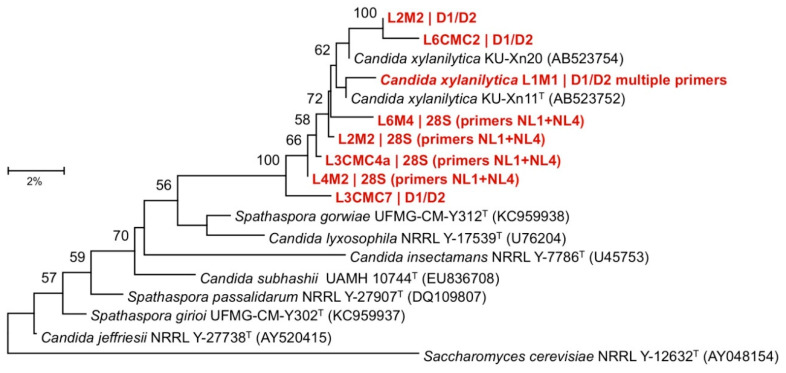
Phylogenetic tree based on D1/D2 region sequence analysis. Samples in red are putative *Candida xylanolytica* strains isolated from larval midguts. The tree was constructed by the neighbor-joining method and *Saccharomyces cerevisiae* was used as the outgroup. Bootstrap values (expressed as percentages of 1000 replications) greater than 50% are shown at branch points. Scale bar represents 2% sequence divergence. T: type strain. Figure Credit: Ming-Ju Chen.

**Table 1 insects-11-00782-t001:** Microbes cultured from the midguts and hindguts of larval *Oryctes rhinoceros*. These microbes were isolated from six larvae whose gut contents were plated into nutrient agar (NA) and carboxymethyl cellulose agar (CMC). Microbe identity given is the closest type specimen in the NCBI RNA database, with sequence similarity in parentheses, except for an Enterobacteriaceae that could not be determined (see results). Microbial density as colony forming units (CFUs) per milligram of midgut or hindgut content is estimated by averaging the CFU/mg of each serial dilution plate with a countable number of colonies of said species, which was only done for NA plates. The ranges varied within an order of magnitude. The number of midgut samples, hindgut samples, and larvae out of six is given. A dash (-) means CFU could not be calculated.

Identity	Agar	Mean CFU/mg (# Samples)		# of Larvae
		Midgut	Hindgut	
*Bacillus cereus* ATCC 14579 (99.72)	NA, CMC	4.7 × 10^5^ (3)	5.5 × 10^5^ (5)	5
*Bacillus proteolyticus* MCCC 1A00365 (99.79)	NA	7.2 × 10^5^ (1)	0	1
*Bacillus toyonensis* BCT-7112 (99.93)	NA	5.0 × 10^4^ (1)	1.5 × 10^7^ (1)	1
** Candida tropicalis* ATCC 750 (99.65)	CMC	- (1)	0	1
** Candida xylanilytica* KU-Xn11T (99.85)	NA, CMC	9.9 × 10^6^ (5)	3.5 × 10^7^ (3)	5
*Citrobacter koseri* LMG 5519 (99.44)	NA, CMC	- (2)	3.1 × 10^8^ (2)	3
*Comamonas nitrativorans* 23310 (99.15)	NA, CMC	- (2)	7.4 × 10^7^ (3)	3
*Diaphorobacter aerolatus* 8604S-37 like (97.38)	CMC	0	- (1)	1
Enterobacteriaceae (97.75)	NA, CMC	3.5 × 10^7^ (3)	1.4 × 10^7^ (3)	3
*Klebsiella pneuomoniae* DSM 30104 (99.44)	NA, CMC	1.1 × 10^6^ (1)	6.8 × 10^7^ (1)	1
*Kluyvera georgiana* ATCC 51603 (99.14)	NA, CMC	3.5 × 10^6^ (1)	3.9 × 10^5^ (1)	1
*Lactococcus lactis* NBRC 100933 like (96.45)	NA	2.6 × 10^6^ (1)	0	1
*Lysinibacillus fusiformis* NBRC 15717 (99.30)	NA, CMC	1.7 × 10^5^ (1)	1.5 × 10^5^ (4)	4
*Pseudomonas citronellolis* NBRC 103043 (98.59)	NA, CMC	0	5.0 × 10^6^ (1)	1
*Staphylococcus epidermidis* NBRC 100911 (99.23)	NA	5.0 × 10^3^ (1)	0	1
*Staphylococcus hominis* GTC 1228 (99.65%)	NA	0	1.7 × 10^6^ (1)	1
*Streptomyces costaricanus* NBRC 100773 or *S. murinus* NBRC 14802 (99.64)	CMC	- (1)	- (1)	1
** Trichoderma virens* DAOM 164916 (99.19)	CMC	- (1)	0	1

* Fungi. # Number.

**Table 2 insects-11-00782-t002:** *Candida xylanilytica* strain cellulase test with tetrazolium blue method. Microbes were reared in nutrient broth, then a sample incubated with CMC in a buffer for 13 min, boiled with tetrazolium blue reagent for 5 min to stop the reaction, and the color change measured using a spectrophotometer. By calibrating the machine with known concentrations of glucose, the concentration (M) of reducing sugars can be quantified. The samples are *Candida xylanilytica* strains from *Oryctes rhinoceros* larvae. The positive control is *Chaetomium globosum*. The negative control is sterile nutrient broth.

Sample	Absorbance at 470 nm	M Sugars
L1M1	1.23	0.1665
L2M2	1.41	0.4398
L3CMC4A	1.19	0.1342
L3CMC7	1.11	0.0871
L4M2	0.99	0.0456
L6M4	1.18	0.1271
L6CMC2	1.36	0.3358
Positive control	1.3	0.2429
Negative control	0.54	0.0040

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
