# Peer review of "Culturing-Enriched Metabarcoding Analysis of the Oryctes rhinoceros Gut Microbiome"

_insects, 2020, doi:10.3390/insects11110782_

Round 1

Reviewer 1 Report

The manuscript presented by Shelomi and Chen is an interesting work were authors studied microbiota associated to the coconut rhinoceros beetle Oryctes rhinoceros.

The manuscript provides interesting information, nonetheless, many data which can guarantee the quality of the provided information are missing. Thus, I strongly suggest waiting until this information is provided, in order to make sure that the published information is not mistaken.

The two major concerns I have are the following:

There is no trustful information on the identification of the bacterial strains: there is no length of the sequence used for comparison with those available in databases, there is no information on the considered most related strain (which should be a type strain of a species, and not any other strain) and there is no information on the percentage of similarity. This information is also missing for the yeasts, but at least the authors performed a phylogenetic tree.

On the other hand, there is no information about the quality of the amplicon sequencing to analyse bacterial diversity through culture-independent methods: we don't know the number of sequences obtained, there is no analysis of the amount of sequences in relation with reaching plateau, etc...

Other comments are included directly in the manuscript.

Regarding the English writing, I am not native speaker and I find the overall manuscript is well written, but the simple summary, which contains several grammar mistakes (some of them highlighted in the file with revisions I am uploading).

In sum, I consider that the manuscript could provide new information, but at the moment it would be dangerous to be published, since it can also provide wrong information. The authors should prove that some data provided are correct, and this is not possible with the manuscript provided in the present form.

Author Response

Dear Reviewer 1,

Thank you for your review and your constructive comments!

We have made the changes recommended on your attached pdf file, and address some of the concerns you mentioned here:

  • Regarding the source of the larvae, they were collected by a collaborator, so we do not know how many trees were involved. These larvae do not live in galleries, but in the pulp of rotten trees, and ours likely all came from one large log. Regardless, they had been living together in the same container of pulp from said log(s) in the lab for a few days before dissection and would have been in close contact together, so we did not expect there to be significant microbiome differences between individuals. Our goal was to find differences between the midgut and hindgut (which we did not find), so collecting multiple individuals from the same location is acceptable, and our sampling sceme still falls within what has been done in previously published works. We updated the methods accordingly.
  • Regarding the lack of other species of Oryctes in Taiwan, we cannot prove a negative, but this region is heavily monitored for invasive pests and has a strong hobbyist beetle collecting community, so a novel Oryctes would likely have been noticed. We modified to text to account for the possibility of an invasion, as requested.
  • We did not have enough samples of fat body or wood pulp for ancom analysis.
  • Family names are not italicized.
  • Regarding the intracellular cellulase testing, we would have liked to do it, but the truth is our lab does not have the biosafety level to study Bacillus cereus, so once we identified it we were obliged not to culture it or study it any further! We intend to analyze these strains in 2022 when we move to a new, more biosecure lab building as part of a larger project on cellulolytic Bacillus cereus, but still think the current research is publishable without those tests.

Regarding the two major concerns from your review:

“There is no trustful information on the identification of the bacterial strains: there is no length of the sequence used for comparison with those available in databases, there is no information on the considered most related strain (which should be a type strain of a species, and not any other strain) and there is no information on the percentage of similarity. This information is also missing for the yeasts, but at least the authors performed a phylogenetic tree.”

We added this information to the methods section and Table 1, except for the lengths, as these varied slightly depending on where we trimmed the 3’ and 5’ ends off the sequences we received from the sequencer. For your reference, all the 16S sequences we used as queries in BLAST were between 1417 to 1435 bp long, which fits given our use of 27F and 1492R primers, while the 18S, 28S, and ITS sequences were shorter.

“On the other hand, there is no information about the quality of the amplicon sequencing to analyse bacterial diversity through culture-independent methods: we don't know the number of sequences obtained, there is no analysis of the amount of sequences in relation with reaching plateau, etc...”

We added this information, and a new supplementary figure showing the plateaus.

With these changes, we hope the revision meets your approval! Thank you again for suggesting these improvements!

Reviewer 2 Report

Well written paper that suits the scope of the journal. The limitations are acknowledged in both the introduction and the discussion. There are few things missing.

  • There is no description of the data analysis procedures in the methods.
  • In the results, authors need to report the full results from the statistics even when differences between groups are not statistically significant.
  • Did you test if the diversity of the microbial community is different between intestine sections?
  • Can you test for differences in cellulase and xylanase activity between your isolates?
  • Section 3.2 needs more quantitative info.

There are also some reviews on metabarcoding and meta-omics in insect science out there that are worth of being cited.

Author Response

We thank the reviewer for their comments. Specific responses are as follows.

“There is no description of the data analysis procedures in the methods.”

We added more such descriptions where appropriate.

“In the results, authors need to report the full results from the statistics even when differences between groups are not statistically significant.”

Which experiments are you referring to? For ancom, that program does not provide full results other than “not significant.” Most of the other studies were qualitative, and also had no statistics to report.

“Did you test if the diversity of the microbial community is different between intestine sections?”

We only compared the midgut and hindgut, and the diversity is not different. We stated this in the results, discussion, and abstract.

“Can you test for differences in cellulase and xylanase activity between your isolates?”

Unfortunately, no. We do not have the biosafety rating to knowingly culture species like Bacillus cereus, so once we identified them we could no longer do any work with them. We will move to a new lab and re-analyze these isolates along with newer ones in a larger project on Bacillus cereus we plan for 2022. For now we feel the manuscript is publishable without these tests.

“Section 3.2 needs more quantitative info.”

Section 3.2 covers a qualitative experiment, not quantitative. The data is presence/absence data, and so does not have statistics.

“There are also some reviews on metabarcoding and meta-omics in insect science out there that are worth of being cited.”

We do not think it necessary to cite reviews on such broad subjects here, but rather cited relevant specific studies (ex: 30-35).

With these changes, we hope the new version meets the reviewer's approval!

Round 2

Reviewer 1 Report

The manuscript has been significantly improved in this revised version.

Nonetheless, I still have two important concerns:

  1. The authors state in the new version of materials and methods the following: "Otherwise, the results were interpreted non-statistically". It is the first time I see something like this in a scientific paper. I do not believe that it is correct to state this. Moreover, I also believe that their analyses with QIIME2 should provide a p-value which allows the authors to interpret the results statistically. 
  2. The rarefaction curve is very strange (it is actually not a curve, but two joined lines).Authors should provide also a rarefaction curve showing the different OTUs against sequencing depth.

Minor comments:

In Table 1, authors have added the names of the strains of each bacterial species, but a T letter after each name indicating when the strain is the type strain of the species should be included (as authors made for the yeast Candida, in which the T appears). Please, ensure that the strain is the type strain and include the T, so readers can quickly know that the comparison is made with a well-classified strain (such as the type strain of the species).

Author Response

We thank the reviewer for their comments, and reply point-by-point as follows.

  • The authors state in the new version of materials and methods the following: "Otherwise, the results were interpreted non-statistically". It is the first time I see something like this in a scientific paper. I do not believe that it is correct to state this. Moreover, I also believe that their analyses with QIIME2 should provide a p-value which allows the authors to interpret the results statistically. 

We have deleted the line. The ancom test QIIME2 runs does not produce p-values. The readout is a volcano plot where, for our data, every point has a W of 0, plus the text "No significant features found." While that is not a p-value, that is statistics, so we added it to the manuscript. No further statistical tests or p-values are required to interpret this data.

  • The rarefaction curve is very strange (it is actually not a curve, but two joined lines).Authors should provide also a rarefaction curve showing the different OTUs against sequencing depth.

The rarefaction curve is the one QIIME2 provided, unedited. The curves can only compare sequencing depth to variables of the sample metadata, not the OTUs.

  • In Table 1, authors have added the names of the strains of each bacterial species, but a T letter after each name indicating when the strain is the type strain of the species should be included (as authors made for the yeast Candida, in which the T appears). Please, ensure that the strain is the type strain and include the T, so readers can quickly know that the comparison is made with a well-classified strain (such as the type strain of the species).

The results of Table 1 come from using BLASTn with the appropriate RNA database while selecting to limit the search database to "Sequences from type material." The top hit strain names were pasted into table 1 unedited. If the type material in the NCBI database does not have a "T" at the end of the strain name, that is NCBI's issue.